# GOT: An Optimal Transport framework for Graph comparison

**Hermina Petric Maretic**
Ecole Polytechnique Fédérale de Lausanne
Signal Processing Laboratory (LTS4)
Lausanne, Switzerland
`hermina.petricmaretic@epfl.ch`

**Mireille EL Gheche**
Ecole Polytechnique Fédérale de Lausanne
Signal Processing Laboratory (LTS4)
Lausanne, Switzerland
`mireille.elgheche@epfl.ch`

**Giovanni Chierchia**
Université Paris-Est, LIGM (UMR 8049)
CNRS, ENPC, ESIEE Paris, UPEM
F-93162, Noisy-le-Grand, France
`giovanni.chierchia@esiee.fr`

**Pascal Frossard**
Ecole Polytechnique Fédérale de Lausanne
Signal Processing Laboratory (LTS4)
Lausanne, Switzerland
`pascal.frossard@epfl.ch`

## Abstract

We present a novel framework based on optimal transport for the challenging problem of comparing graphs. Specifically, we exploit the probabilistic distribution of smooth graph signals defined with respect to the graph topology. This allows us to derive an explicit expression of the Wasserstein distance between graph signal distributions in terms of the graph Laplacian matrices. This leads to a structurally meaningful measure for comparing graphs, which is able to take into account the global structure of graphs, while most other measures merely observe local changes independently. Our measure is then used for formulating a new graph alignment problem, whose objective is to estimate the permutation that minimizes the distance between two graphs. We further propose an efficient stochastic algorithm based on Bayesian exploration to accommodate for the non-convexity of the graph alignment problem. We finally demonstrate the performance of our novel framework on different tasks like graph alignment, graph classification and graph signal prediction, and we show that our method leads to significant improvement with respect to the state-of-art algorithms.

## 1 Introduction

With the rapid development of digitisation in various domains, the volume of data increases very rapidly, with many of those taking the form of structured data. Such information is often represented by graphs that capture potentially complex structures. It stays however pretty challenging to analyse, classify or predict graph data, due to the lack of efficient measures for comparing graphs. In particular, the mere comparison of graph matrices is not necessarily a meaningful distance, as different edges can have a diverse importance in the graph. Spectral distances have also been proposed [1, 2], but they usually do not take into account all the information provided by the graphs, focusing only on the Laplacian matrix eigenvectors and ignoring a large portion of the structure encoded in eigenvectors. In addition to the lack of effective distances, a major difficulty with graph representations is that their nodes may not be aligned, which further complicates graph comparisons.

In this paper, we propose a new framework for graph comparison, which permits to compute both the distance between two graphs under unknown permutations, and the transportation plan for data from one graph to another, under the assumption that the graphs have the same number of vertices. Instead

of comparing graph matrices directly, we propose to look at the smooth graph signal distributions associated to each graph, and to relate the distance between graphs to the distance between the graph signal distributions. We resort to optimal transport for computing the Wasserstein distance between distributions, as well as the corresponding transportation plan. Optimal transport (OT) was introduced by Monge [3], and reformulated in a more tractable way by Kantorovich [4]. It has been a topic of great interest both theoretically and practically [5], and has recently been largely revisited with new applications in image processing, data analysis, and machine learning [6]. Interestingly, the Wasserstein distance takes a closed-form expression in our settings, which essentially depends on the Laplacian matrices of the graphs under comparison. We further show that the Wasserstein distance has the important advantage of capturing the main structural information of the graphs.

Equipped with this distance, we formulate a new graph alignment problem for finding the permutation that minimises the mass transportation between a "fixed" distribution and a "permuted" distribution. This yields a nonconvex optimization problem that we solve efficiently with a novel stochastic gradient descent algorithm. It permits to efficiently align and compare graphs, and it outputs a structurally meaningful distance and transport map. These are important elements in graph analysis, comparison, or graph signal prediction tasks. We finally illustrate the benefits of our new graph comparison framework in representative tasks such as noisy graph alignment, graph classification, and graph signal transfer. Our results show that the proposed distance outperforms both Gromov-Wasserstein and Euclidean distance for what concerns the graph alignment and graph clustering. In addition, we show the use of transport maps to predict graph signals. To the best of our knowledge, this is the only framework for graph comparison that includes the possibility to adapt graph signals to another graph.

## 1.1 Related work

In the literature, many methods have formulated the graph matching as a quadratic assignment problem [7, 8], under the constraint that the solution is a permutation matrix. As this is an NP-hard problem, different relaxations have been proposed to find approximate solutions. In this context, spectral clustering [9, 10] emerged as a simple relaxation, which consists of finding the orthogonal matrix whose squared entries sum to one, but the drawback is that the matching accuracy is suboptimal. To improve on this behavior, the semi-definite programming relaxation was adopted to tackle the graph matching problem by relaxing the non-convex constraint into a semi-definite one [11]. Spectral properties have also been used to inspect graphs and define different classes of graphs for which the convex relaxation is equivalent to the original graph maching problem [12] [13]. Other works focus on the general problem and propose provably tight convex relaxations for all graph classes [14]. Based on the assumption that the space of doubly-stochastic matrices is a convex hull, the graph matching problem was relaxed into a non-convex quadratic problem in [15, 16]. A related approach was recently proposed to approximate discrete graph matching in the continuous domain asymptotically by using separable functions [17]. Along similar lines, a Gumbel-sinkhorn network was proposed to infer permutations from data [18, 19]. The approach consists of producing a discrete permutation from a continuous doubly-stochastic matrix obtained with the Sinkhorn operator.

Closer to our framework, some recent works have studied the graph alignment problem from an optimal transport perspective. For example, Flamary *et al.* [20] propose a method based on optimal transport for empirical distributions with a graph-based regularization. The objective of this work is to compute an optimal transportation plan by controlling the displacement of a pair of points. Graph-based regularization encodes neighborhood similarity between samples on either the final position of the transported samples, or their displacement [21]. Gu *et al.* [22] define a spectral distance by assigning a probability measure to the nodes via the spectrum representation of each graph, and by using Wasserstein distances between probability measures. This approach however does not take into account the full graph structure in the alignment problem. Nikolentzos *et al.* [23] proposed instead to match the graph embeddings, where the latter are represented as bags of vectors, and the Wasserstein distance is computed between them. The authors also propose a heuristic to take into account possible node labels or signals.

Another line of works have looked at more specific graphs. Memoli [24] investigates the Gromov-Wasserstein distance for object matching, and Peyré *et al.* [25] propose an efficient algorithm to compute the Gromov-Wasserstein distance and the barycenter of pairwise dissimilarity matrices. The algorithm uses entropic regularization and Sinkhorn projections, as proposed by [26]. The work has many interesting applications, including multimedia with point-cloud averaging and matching, but

also natural language processing with alignment of word embedding spaces [27]. Vayer *et al.* [28] build on this work and propose a distance for graphs and signals living on them. The problem is given as a combination between the Gromov-Wasserstein of graph distance matrices and the Wasserstein distance of graph signals. However, while the above methods solve the alignment problem using optimal transport, the simple distances between aligned graphs do not take into account its global structure and the methods do not consider the transportation of signals between graphs.

## 1.2 Organization

In this paper, we propose to resort to smooth graph signal distributions in order to compare graphs, and develop an effective algorithm to align graphs under a priori unknown permutations. The paper is organized as follows. Section 2 details the graph alignment with optimal transport. Section 3 presents the algorithm for solving the proposed approach via a stochastic gradient technique. Section 4 provides an experimental validation of graph matching in the context of graph classification, and graph signal transfer. Finally, the conclusion is given in Section 5.

## 2 Graph Alignment with Optimal Transport

Despite recent advances in the analysis of graph data, it stays pretty challenging to define a meaningful distance between graphs. Even more, a major difficulty with graph representations is the lack of node alignment, which prevents from performing direct quantitative comparisons between graphs. In this section, we propose a new distance based on Optimal Transport (OT) to compare graphs through smooth graph signal distributions. Then, we use this distance to formulate a new graph alignment problem, which aims at finding the permutation matrix that minimizes the distance between graphs.

### 2.1 Preliminaries

We denote by $\mathcal{G} = (V, E)$ a graph with a set $V$ of $N$ vertices and a set $E$ of edges. The graph is assumed to be connected, undirected, and edge weighted. The adjacency matrix is denoted by $W \in \mathbb{R}^{N \times N}$. The degree of a vertex $i \in V$, denoted by $d(i)$, is the sum of weights of all the edges incident to $i$ in the graph $\mathcal{G}$. The degree matrix $D \in \mathbb{R}^{N \times N}$ is then defined as:

$$D_{i,j} = \begin{cases} d(i) & \text{if } i = j \\ 0 & \text{otherwise.} \end{cases} \tag{1}$$

Based on $W$ and $D$, the Laplacian matrix of $\mathcal{G}$ is

$$L = D - W. \tag{2}$$

Moreover, we consider additional attributes modelled as features on the graph vertices. Assuming that each node is associated to a scalar feature, the graph signal takes the form of a vector in $\mathbb{R}^N$.

### 2.2 Smooth graph signals

Following [29], we interpret graphs as key elements that drive the probability distributions of signals. Specifically, we consider two graphs $\mathcal{G}_1$ and $\mathcal{G}_2$ with Laplacian matrices $L_1$ and $L_2$, and we consider signals that follow the normal distributions with zero mean and $L_1^\dagger$ or $L_2^\dagger$ as covariance matrix [30], namely[1]

$$\nu^{\mathcal{G}_1} = \mathcal{N}(0, L_1^\dagger) \tag{3}$$

$$\mu^{\mathcal{G}_2} = \mathcal{N}(0, L_2^\dagger). \tag{4}$$

The above formulation means that the graph signal values vary slowly between strongly connected nodes [30]. This assumption is verified for most common graph and network datasets. It is further used in many graph inference algorithms implicitly representing a graph through its smooth signals [31–33]. Furthermore, the smoothness assumption is used as regularization in many graph applications, such as robust principal component analysis [34] and label propagation [35].

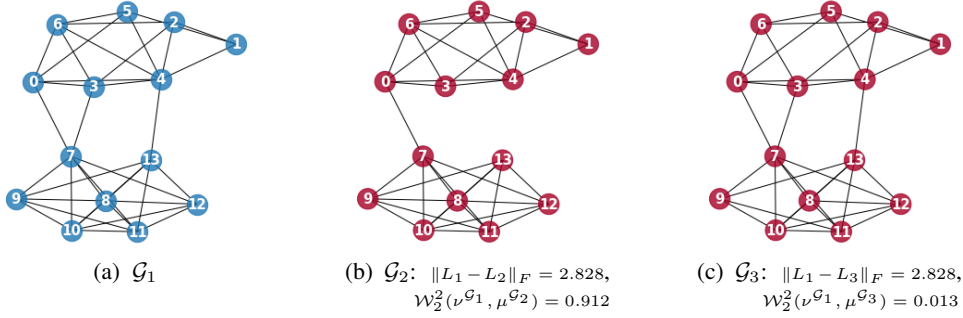

(a) $\mathcal{G}_1$

(b) $\mathcal{G}_2$: $\|L_1 - L_2\|_F = 2.828$, $\mathcal{W}_2^2(\nu^{\mathcal{G}_1}, \mu^{\mathcal{G}_2}) = 0.912$

(c) $\mathcal{G}_3$: $\|L_1 - L_3\|_F = 2.828$, $\mathcal{W}_2^2(\nu^{\mathcal{G}_1}, \mu^{\mathcal{G}_3}) = 0.013$

Figure 1: Illustration of the structural differences captured with Wasserstein distance between graphs defined in (5). The graphs $\mathcal{G}_2$ and $\mathcal{G}_3$ are both copies of $\mathcal{G}_1$, with 2 edges removed. The modification in $\mathcal{G}_2$ is very influential, as the two communities are almost disconnected; here, both Frobenius norm and Wasserstein distance measure a significant difference w.r.t. $\mathcal{G}_1$. Conversely, the modification in $\mathcal{G}_3$ is hardly noticeable; here, the Frobenius norm still measures a significant difference, whereas the Wasserstein distance does not. The latter is a desirable property in the context of graph comparison.

## 2.3 Wasserstein distance between graphs

Instead of comparing graphs directly, we propose to look at the signal distributions, which are governed by the graphs. Specifically, we measure the dissimilarity between two aligned graphs $\mathcal{G}_1$ and $\mathcal{G}_2$ through the Wasserstein distance of the respective distributions $\nu^{\mathcal{G}_1}$ and $\mu^{\mathcal{G}_2}$. More precisely, the 2-Wasserstein distance corresponds to the minimal "effort" required to transport one probability measure to another with respect to the Euclidean norm [3], that is

$$W_2^2(\nu^{\mathcal{G}_1}, \mu^{\mathcal{G}_2}) = \inf_{T_\#\nu^{\mathcal{G}_1}=\mu^{\mathcal{G}_2}} \int_{\mathcal{X}} \|x - T(x)\|^2 \, d\nu^{\mathcal{G}_1}(x), \tag{5}$$

where $T_\#\nu^{\mathcal{G}_1}$ denotes the push forward of $\nu^{\mathcal{G}_1}$ by the transport map $T\colon \mathcal{X} \to \mathcal{X}$ defined on a metric space $\mathcal{X}$. For normal distributions such as $\nu^{\mathcal{G}_1}$ and $\mu^{\mathcal{G}_2}$, the 2-Wasserstein distance can be explicitly written in terms of their covariance matrices [36], yielding

$$W_2^2(\nu^{\mathcal{G}_1}, \mu^{\mathcal{G}_2}) = \mathrm{Tr}\left(L_1^\dagger + L_2^\dagger\right) - 2\,\mathrm{Tr}\left(\sqrt{L_1^{\frac{1}{2}} L_2^\dagger L_1^{\frac{1}{2}}}\right), \tag{6}$$

and the optimal transportation map is $T(x) = L_1^{\frac{\dagger}{2}} \left(L_1^{\frac{\dagger}{2}} L_2^\dagger L_1^{\frac{\dagger}{2}}\right)^{\frac{\dagger}{2}} L_1^{\frac{\dagger}{2}} x$.

The Wasserstein distance captures the structural information of the graphs under comparison. It is sensitive to differences that cause a global change in the connection between graph components, while it gives less importance to differences that have a small impact on the whole graph structure. Indeed, as graphs are represented through the distribution of smooth signals, the Wasserstein distance essentially measures the discrepancy in lower graph frequencies, known to capture the global graph structure. This behaviour is illustrated in Figure 1 by a comparison with a simple distance that is the Euclidean norm between the Laplacian matrices of the graphs.[2]

Moreover, the optimal transportation map enables the movement of signals from one graph to another. This is a continuous Lipshitz mapping that adapts a graph signal to the distribution of another graph, while keeping similarity. This results in a simple and efficient prediction of the signal on another graph. Clearly, signals that are more likely in the observed distribution will have a more robust transportation, and different Gaussian signal models (in Equations 3 and 4) might be more appropriate for non-smooth signals [37].

## 2.4 Graph alignment

Equiped with a measure to compare aligned graphs =of the same size through signal distributions, we now propose a new formulation of the graph alignment problem. It is important to note that the graph

**Algorithm 1** Approximate solution to the graph alignment problem defined in (8).

---

**Require:** Graphs $\mathcal{G}_1$ and $\mathcal{G}_2$
**Require:** Sampling size $S \in \mathbb{N}$, learning rate $\gamma > 0$, and constant $\tau > 0$
**Require:** Random initialization of matrices $\eta_0$ and $\sigma_0$
 1: **for** $t = 0, 1, \ldots$ **do**
 2:     Draw samples $\{\epsilon_t^{(s)}\}_{1 \leq s \leq S}$ from the distribution $q_{\text{unit}}$
 3:     Define the stochastic approximation of the cost function as

$$J_t(\eta_t, \sigma_t) = \frac{1}{S} \sum_{s=1}^{S} \mathcal{W}_2^2 \left( \nu^{\mathcal{G}_1}, \mu^{\mathcal{G}_2}_{\mathcal{S}_\tau(\eta_t + \sigma_t \odot \epsilon_t^{(s)})} \right)$$

 4:     $g_t \leftarrow$ gradient of $J_t$ evaluated at $(\eta_t, \sigma_t)$
 5:     $(\eta_{t+1}, \sigma_{t+1}) \leftarrow$ update of $(\eta_t, \sigma_t)$ using $g_t$
 6: **return** $P = \mathcal{S}_\tau(\eta_*)$

---

signal distributions depend on the enumeration of nodes chosen to build $L_1$ and $L_2$. While in some cases (e.g., dynamically changing graphs, multilayer graphs, etc. . . ) a consistent enumeration can be trivially chosen for all graphs, it generally leads to the challenging problem of estimating an a priori unknown permutation between graphs. In our approach, we are given two connected graphs $\mathcal{G}_1$ and $\mathcal{G}_2$, each with $N$ distinct vertices and with different sets of edges. We aim at finding the optimal transportation map $T$ from $\mathcal{G}_1$ to $\mathcal{G}_2$. However, the vertices of these graphs are not necessarily aligned. In order to take all possible enumerations into account, we define the probability measure of a permuted representation for the graph $\mathcal{G}_2$ as

$$\mu_P^{\mathcal{G}_2} = \mathcal{N}\left(0, (P^\top L_2 P)^\dagger\right) = \mathcal{N}(0, P^\top L_2^\dagger P), \tag{7}$$

where $P \in \mathbb{R}^{N \times N}$ is a permutation matrix. Consequently, our graph alignment problem consists in finding the permutation that minimizes the mass transportation between $\nu^{\mathcal{G}_1}$ and $\mu_P^{\mathcal{G}_2}$, which reads

$$\underset{P \in \mathbb{R}^{N \times N}}{\text{minimize}} \ \mathcal{W}_2^2\left(\nu^{\mathcal{G}_1}, \mu_P^{\mathcal{G}_2}\right) \qquad \text{s.t.} \qquad \begin{cases} P \in [0,1]^N \\ P \mathbb{1}_N = \mathbb{1}_N \\ \mathbb{1}_N^\top P = \mathbb{1}_N \\ P^\top P = \mathrm{I}_{N \times N}, \end{cases} \tag{8}$$

where $\mathbb{1}_N = [1 \ \ldots \ 1]^\top \in \mathbb{R}^N$ and $\mathrm{I}_{N \times N}$ is the $N \times N$ identity matrix. According to (3), (6), (7), the above distance boils down to

$$\mathcal{W}_2^2\left(\nu^{\mathcal{G}_1}, \mu_P^{\mathcal{G}_2}\right) = \mathrm{Tr}\left(L_1^\dagger + P^T L_2^\dagger P\right) - 2\,\mathrm{Tr}\left(\sqrt{L_1^{\frac{\dagger}{2}} P^T L_2^\dagger P L_1^{\frac{\dagger}{2}}}\right). \tag{9}$$

The optimal permutation allows us to compare $\mathcal{G}_1$ and $\mathcal{G}_2$ when the consistent enumeration of nodes is not available. This is however a non-convex optimization problem that cannot be easily solved with standard tools. In the next section, we present an efficient algorithm to tackle this problem.

## 3 GOT Algorithm

We propose to solve the OT-based graph alignment problem described in the previous section via stochastic gradient descent. The latter is summarized in Algorithm 1, and its derivation is presented in the remaining of this section.

### 3.1 Optimization

The main difficulty in solving Problem (8) arises from the constraint that $P$ is a permutation matrix, since it leads to a discrete optimization problem with a factorial number of feasible solutions. We propose to circumvent this issue through an implicit constraint reformulation. The idea is that the constraints in (8) can be enforced implicitly by using the Sinkhorn operator [38, 26, 39, 18]. Given a square matrix $P \in \mathbb{R}^{N \times N}$ (not necessarily a permutation) and a small constant $\tau > 0$, the Sinkhorn

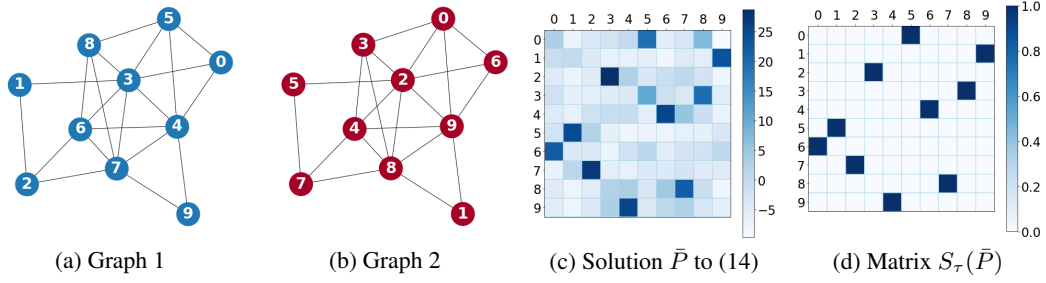

| (a) Graph 1 | (b) Graph 2 | (c) Solution $\bar{P}$ to (14) | (d) Matrix $S_\tau(\bar{P})$ |
|---|---|---|---|

Figure 2: Illustrative example of the graph alignment problem. The solution to (14) is a matrix $\bar{P}$ whose rows may be interpreted as assignment log-likelihoods. Applying the Sinkhorn operator to $\bar{P}$ yields a matrix whose rows are assignment probabilities from Graph 1 (columns) to Graph 2 (rows).

operator $\mathcal{S}_\tau$ normalizes the rows and columns of $\exp(P/\tau)$ via the multiplication by two diagonal matrices $A$ and $B$, yielding[3]

$$\mathcal{S}_\tau(P) = A \exp(P/\tau) B. \tag{10}$$

The diagonal matrices $A$ and $B$ are computed iteratively as follows:

$$A^{[k]} = \operatorname{diag}\left(P^{[k]} \mathbb{1}_N\right)^{-1} \tag{11}$$

$$B^{[k]} = \operatorname{diag}\left(\mathbb{1}_N^\top A^{[k]} P^{[k]}\right)^{-1} \tag{12}$$

$$P^{[k+1]} = A^{[k]} P^{[k]} B^{[k]}, \tag{13}$$

with $P^{[0]} = \exp(P/\tau)$. It can be shown [18] that the operator $\mathcal{S}_\tau$ yields a permutation matrix in the limit $\tau \to 0$. Consequently, with a slight abuse of notation (as $P$ no longer denotes a permutation), we can rewrite Problem (8) as follows

$$\underset{P \in \mathbb{R}^{N \times N}}{\text{minimize}} \ \mathcal{W}_2^2\left(\nu^{\mathcal{G}_1}, \mu_{\mathcal{S}_\tau(P)}^{\mathcal{G}_2}\right). \tag{14}$$

The above cost function is differentiable [40], and can be thus optimized by gradient descent. An illustrative example of a solution of the proposed approach is presented in Fig. 2.

## 3.2 Stochastic exploration

Problem (14) is highly nonconvex, which may cause gradient descent to converge towards a local minimum. Hence, instead of directly optimizing the cost function in (14), we can optimize its expectation w.r.t. the parameters $\theta$ of some distribution $q_\theta$, yielding

$$\underset{\theta}{\text{minimize}} \ \mathbb{E}_{P \sim q_\theta}\left\{\mathcal{W}_2^2\left(\nu^{\mathcal{G}_1}, \mu_{\mathcal{S}_\tau(P)}^{\mathcal{G}_2}\right)\right\}. \tag{15}$$

The optimization of the expectation w.r.t. the parameters $\theta$ aims at shaping the distribution $q_\theta$ so as to put all its mass on a minimizer of the original cost function, thus integrating the use of Bayesian exploration in the optimization process.

A standard choice for $q_\theta$ in continuous optimization is the multivariate normal distribution, thus leading to $\theta = (\eta, \sigma) \in \mathbb{R}^{N \times N} \times \mathbb{R}^{N \times N}$ and $q_\theta = \prod_{i,j} \mathcal{N}\left(\eta_{ij}, \sigma_{ij}^2\right)$. By leveraging the reparameterization trick [41, 42], which boils down to the equivalence

$$\left(\forall (i,j) \in \{1, \ldots, N\}^2\right) \quad P_{ij} \sim \mathcal{N}\left(\eta_{ij}, \sigma_{ij}^2\right) \quad \Leftrightarrow \quad \begin{cases} \epsilon_{ij} \sim \mathcal{N}(0, 1) \\ P_{ij} = \eta_{ij} + \sigma_{ij}\epsilon_{ij}, \end{cases} \tag{16}$$

the above problem can be reformulated as[4]

$$\underset{\eta, \sigma}{\text{minimize}} \ \underbrace{\mathbb{E}_{\epsilon \sim q_{\text{unit}}}\left\{\mathcal{W}_2^2\left(\nu^{\mathcal{G}_1}, \mu_{\mathcal{S}_\tau(\eta + \sigma \odot \epsilon)}^{\mathcal{G}_2}\right)\right\}}_{J(\eta, \sigma)}, \tag{17}$$

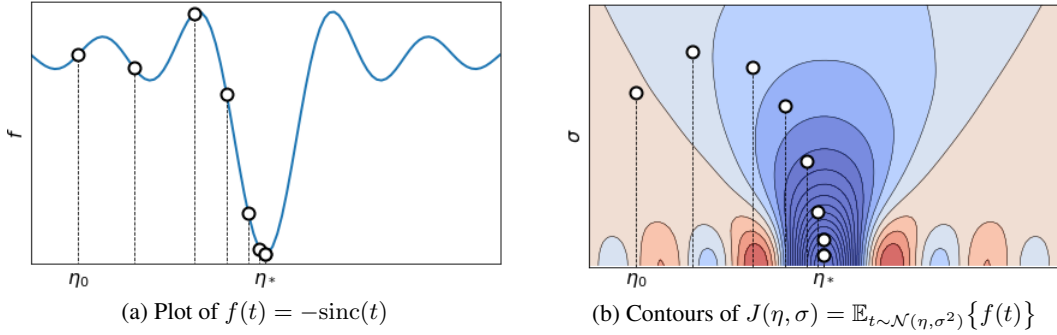

(a) Plot of $f(t) = -\mathrm{sinc}(t)$        (b) Contours of $J(\eta, \sigma) = \mathbb{E}_{t \sim \mathcal{N}(\eta, \sigma^2)}\{f(t)\}$

Figure 3: Illustrative example of stochastic exploration. The white circles mark the iterates $(\eta_0, \sigma_0), \ldots, (\eta_*, \sigma_*)$ produced by optimizing $J$ (the expectation of $f$) via stochastic gradient descent. As this optimization is performed in the space of parameters $\eta$ and $\sigma$ (see the right panel), the algorithm avoids local minima and successfully converges to the global minimum of both $J$ and $f$.

where $q_{\mathrm{unit}} = \prod_{i,j} \mathcal{N}(0, 1)$ denotes the multivariate normal distribution with zero mean and unitary variance. The advantage of this reformulation is that the gradient of the above stochastic function can be approximated by sampling from the parameterless distribution $q_{\mathrm{unit}}$, yielding

$$\nabla J(\eta, \sigma) \approx \sum_{\epsilon \sim q_{\mathrm{unit}}} \nabla \mathcal{W}_2^2(\nu^{\mathcal{G}_1}, \mu^{\mathcal{G}_2}_{\mathcal{S}_\tau(\eta + \sigma \odot \epsilon)}). \tag{18}$$

The problem can be thus solved by stochastic gradient descent [43]. An illustrative application of this approach on a simple one-dimensional nonconvex function is presented in Fig. 3. Under mild assumptions, the algorithm converges almost surely to a critical point, which is not guaranteed to be the global minimum, as the problem is nonconvex.

The computational complexity of the naive implementation is $O(N^3)$ per iteration, due to the matrix square-root operation based on a singular value decomposition (SVD). A better option consists of approximating the matrix square-root with the Newton's method [44]. These iterations only involve matrix multiplications, which can take advantage of the matrix sparsity, thus resulting in a faster implementation than SVD. Moreover, the computation of pseudo-inverses can be avoided by adding a small diagonal shift to the Laplacian matrices and directly computing the inverse matrices, which is orders of magnitude faster. This is not a large concern though, as it can be done in preprocessing and only needs to be done once. Finally, the algorithm was implemented using automatic differentiation (in PyTorch with AMSGrad [45]).

## 4 Experimental results

We illustrate the behaviour of our approach, named GOT, in terms of both distance metric computation and transportation map inference. We show how, due to the ability of our distance metric to strongly capture structural properties, it can be beneficial in computing alignment between structured graphs even when they are very different. For similar reasons, the metric is able to properly separate instances of random graphs according to their original model. Finally, we show illustrations of the use of transportation maps for signal prediction in simple image classes.

Prior to running experiments, we chose the parameters $\tau$ (Sinkhorn) and $\gamma$ (learning rate) with grid search, while $S$ (sampling size) was fixed empirically. In all experiments, we set $\tau = 5, \gamma = 0.2$, and $S = 30$. We set the maximal number of Sinkhorn iterations to 10, and we run stochastic gradient descent for 3000 iterations (even though the algorithm converges long before, after around 1000 iterations, typically). As our algorithm seems robust to different initialisation, we used random initialisation in all our experiments. The code is available at `https://github.com/Hermina/GOT`.

### 4.1 Alignment of structured graphs

We generate a stochastic block model graph with 40 nodes and 4 communities. A noisy version of this graph is created by randomly removing edges within communities with probability $p = 0.5$, and

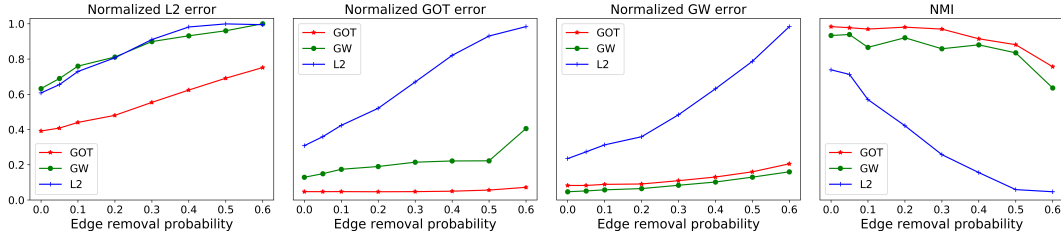

Figure 4: Alignment and community detection performance for distorted stochastic block model graphs as a function of the edge removal probability. The first three plots show different error measures (closer to 0 the better); the last one shows the community detection performance in terms of Normalized Mutual Information (NMI closer to 1 the better).

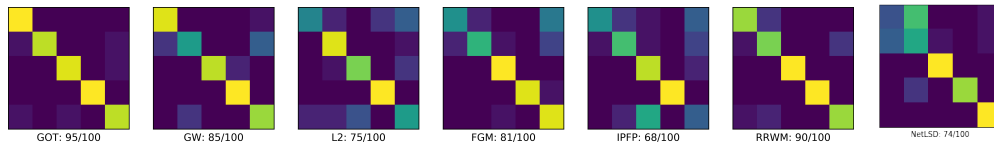

Figure 5: Confusion matrices for 1-NN classification results on random graph models. Rows represent actual classes, while columns are predicted classes: SBM2, SBM3, RG, BA, WS respectively.

edges between communities with increasing probabilities $p \in [0, 0.6]$. We then generate a random permutation to change the order of nodes in the noisy graph. We investigate the influence of a distance metric on alignment recovery. We compare three different methods for graph alignment, namely the proposed method based on the suggested Wasserstein distance between graphs (GOT), the proposed stochastic algorithm with the Euclidean distance (L2), and the state-of-the-art Gromov-Wasserstein distance [25] [28] for graphs (GW), based on the Euclidean distance between shortest path matrices, as proposed in [28]. We repeat each experiment 50 times, after adjusting parameters for all compared methods, and show the results in Figure 4.

Apart from analysing the distance between aligned graphs with all three error measures, we also evaluate the structural recovery of these community-based models by inspecting the normalized mutual information (NMI) for community detection. While GW slightly outperforms GOT in terms of its own error measure, GOT clearly performs better in terms of all other inspected metrics. In particular, the last plot shows that the structural information is well captured in GOT, and communities are successfully recovered even when the graphs contain a large amount of introduced perturbations.

## 4.2 Graph classification

We tackle the task of graph classification on random graph models. We create 100 graphs following five different models (20 per model), namely Stochastic Block Model [46] with 2 blocks (SBM2), Stochastic Block Model with 3 blocks (SBM3), random regular graph (RG) [47], Barabasy-Albert model (BA) [48], and Watts-Strogatz model (WS) [49]. All graphs have 20 nodes and a similar number of edges to make the task more meaningful, and are randomly permuted. We use GOT to align graphs, and eventually use a simple non-parametric 1-NN classification algorithm to classify graphs. We compare to several methods for graph alignment: GW [25, 28], FGM [50], IPFP [51], RRWM [15] and NetLSD[52]. We present the results in terms of confusion matrices in Figure 5, accompanied with their accuracy scores. GOT clearly outperforms the other methods in terms of general accuracy, with GW and RRWM also performing well, but having more difficulties with SBMs and the WS model. This once again suggests that GOT is able to capture structural information of graphs.

## 4.3 Graph signal transportation

Finally, we look at the relevance of the transportation plans produced by GOT in illustrative experiments with simple images. We use the MNIST dataset, which contains around 60000 images of size

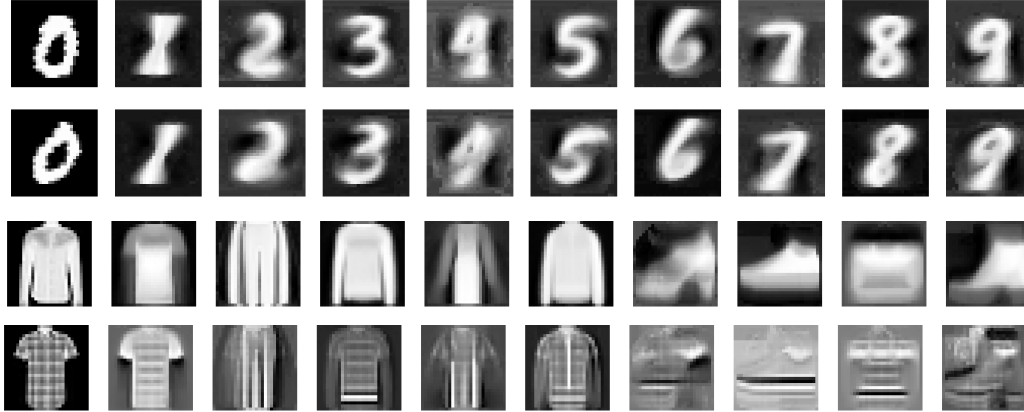

Figure 6: *First two rows:* Original "zero" digits in MNIST dataset, and their images transported to graphs of different digits. The transported digits in each row follow the inclination of the original zero digit. *Last two rows:* Original "Shirt" images in Fashion MNIST dataset, and their images transported to the graphs of other classes ("T-shirt", "Trouser", "Pullover", "Dress", "Coat", "Sandal", "Sneaker","Bag", "Ankle boot").

$28 \times 28$ displaying handwritten digits from 0 to 9, with 6000 per class. For each class $c \in \{0, \dots, 9\}$, we stack all the available images into a feature matrix of size $6000 \times 784$, and we build a graph over the resulting 784 feature vectors. To construct a graph, we first create a 20-nearest-neighbour binary graph, which we then square (multiply with itself) to obtain the final graph, capturing 2-hop distances and creating more meaningful weights. Hence, each class of digits is represented by a graph of 784 nodes (i.e., image pixels), yielding 9 aligned graphs $\mathcal{G}_{\mathrm{zero}}, \mathcal{G}_{\mathrm{one}}, \dots, \mathcal{G}_{\mathrm{nine}}$.

Each image of a given class can be seen as a smooth signal $x \in \mathbb{R}^{784}$ that lives on the corresponding graph. A transportation plan $T$ is then constructed between the source graph (e.g., $\mathcal{G}_{\mathrm{zero}}$) and all other graphs (e.g., $\mathcal{G}_{\mathrm{one}}, \mathcal{G}_{\mathrm{two}}, \dots, \mathcal{G}_{\mathrm{nine}}$). Figure 6 shows two original "zero" digits with different inclination, transported to the graphs of all other digits. We can see that the predicted digits are recognisable, because they are adapted to their corresponding graphs, and they further keep the similarity with the original digit in terms of inclination.

We repeated the same experiment on Fashion MNIST, and reported the results in Figure 6. By transporting a "Shirt" image to the graphs of classes "T-shirt", "Trouser", "Pullover", "Dress", "Coat", "Sandal", "Sneaker", "Bag", "Ankle boot", we can remark that the predicted images are still recognisable with a good degree of fidelity. Furthermore, we observe that the white shirt translates to white clothing items, while the textured shirt leads to textured items. This experiment confirms the potential of GOT in graph signal prediction through adaptation of a graph signal to another graph.

## 5 Conclusion

We presented an optimal transport based approach for computing the distance between two graphs and the associated transportation plan. Equipped with this distance, we formulated the problem of finding the permutation between two unaligned graphs, and we proposed to solve it with a novel stochastic gradient descent algorithm. We evaluated the proposed approach in the context of graph alignment, graph classification, and graph signal transportation. Our experiments confirmed that GOT can efficiently capture the structural information of graphs, and the proposed transportation plan leads to promising results for the transfer of signals from one graph to another.

## 6 Acknowledgment

Giovanni Chierchia was supported by the CNRS INS2I JCJC project under grant 2019OSCI.

## Footnotes

[1]Note that $\dagger$ denotes a pseudoinverse operator.

[2]Note that in our setting a possible alternative to the Wasserstein distance could be the Kullback-Leibler (KL) divergence, whose expression is explicit for normal distributions.

[3]Note that $\exp$ is applied element-wise to ensure the positivity of the matrix entries.

[4]Note that $\odot$ is the entry-wise (Hadamard) product between matrices.

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
