[Reviews · NeurIPS 2019]

Reviewer 1



This paper presents a novel approach for computing a distance between (unaligned) graphs using the Wasserstein distance between signals (or, more specifically, random Gaussian vectors) on the graphs. The graph alignment problem is then solved through the minimization of the Wasserstein distance (which has an explicit formulation in this case) over the set of permutation matrices, encoded using the Sinkhorn operator. The paper is easy to read and very interesting. The ideas proposed for this challenging problem (the computation of graph distances and graph alignment) are very elegant and, based on the experimental section, efficient in practice. The use of the Wasserstein distance between graph signals for computing graph distances appears novel, and the work contains many interesting ideas that would be valuable for a researcher attending the conference and working on a similar topic. My two main concerns are: 1) Weak experimental section: Several technical aspects of the algorithm are missing. For example, the choice of parameter \tau, number of samples S and number of iterations of the Sinkhorn algorithm. Are you using automatic differentiation to perform the stochastic gradient descent? How robust is the optimization algorithm to different initialization? In practice, do you run the optimization multiple times to avoid local minimums? The computation time and complexity of the algorithm would also make a valuable addition to this work, as it is hard to evaluate the speed of such a computation based on the text, and the algorithm seems intuitively rather slow due to the many iterations of the Sinkhorn operator required to converge to the permutation matrix. Is this number of iterations large in practice? 2) The distance is only applicable for graphs of the same size. This should be clearly stated in the abstract, as many graph datasets contain graphs of multiple sizes, making the approach impossible to apply. Is the method easily extendable to graphs of different sizes? Minor comments and typos: 1) In the abstract, the authors mention that the distance can capture the "global structure of graphs", although this claim is never proved nor motivated. Can you elaborate on this aspect? 2) The notation N(0,x) is ambiguous, as x may represent the variance or the standard deviation. Can you clarify this before or after eq.(3)? 3) l.15: "the-state" -> "the state" 4) l.53: "a NP-hard" -> "an NP-hard" 5) footnote 2: "ensures" -> "ensure"

Reviewer 2



In this paper the authors tackle the graph alignment problem (and other related problems) with tools from the optimal transport area. Both the problem and the tools are of great significance. The paper is very well written, it's very pedagogical and the description and examples (Figs 1,2,3) are very illustrative. It's definitively a very nice paper to read. In the introduction, the first paragraph reviewing graph alignment methods may be complemented with very important works on this direction. For instance: - On convex relaxation of graph isomorphism, PNAS, Aflalo Y., Bronstein A., Kimmel R. - A Path Following Algorithm for the Graph Matching Problem, PAMI, Zaslavskiy, M., Bach, F. & Vert, J. - On spectral properties for graph matching and graph isomorphism problems, IMAIAI, Fiori M., Sapiro G. - DS++: A flexible, scalable and provably tight relaxation for matching problems, Dym N., Maron H., Lipman Y. This is a minor point: I understand that the use of the Sinkhorn operator it's convenient in the formulation of the optimization problem due to its differentiability. However, after getting the matrix \bar{P}, which in general is not a permutation matrix, other methods may be used to obtain a permutation matrix. From the projection to the permutation set, which is a LAP, to more complex methods like some in the cited papers above. I'm not sure if this could improve the results. In particular, in the example of Fig. 2, I'm sure it would give the same (correct) answer. For directed graphs, the problem is the use of the Laplacian (and its pseudoinvesere)? I guess that the optimization requires the pseudo-inverse of the Laplacians. How does this scale with the size of the graph? And how is the convergence of the algorithm? (both in speed and when does it fail) I would have liked to see an experiment analyzing the performance of graph alignment but for inexact graph matching. There's also a small error in Figure 1 (b) and (c). According to the numbers presented, the norm there is the Frobenius norm, and not the 2 norm.

Reviewer 3



The work is original (as far as I know). I had never seen the use of OT that way to propose a distance between graphs written as eq. (6) and then (9) when a permutation is involved; despite the fact that the result is a direct application of known results) and significant. The whole manuscript shows that this way of writing a distance between graphs, that gives also a transportation plan from one graph to another, can use in practical situations for graph alignement, a known and important problem. I don't have much comments on the work, as I think it is a very good and that it is already very clear. Some improvements could always be made, as described in a next Section. The authors' feedback after reviews is very good also, conforting my very good impression about this work.

[Author Response · NeurIPS 2019]

We thank the reviewers for their thorough reviews and useful suggestions. We address below the main concerns raised by the reviewers. These will be incorporated in the final version, along with corrections of minor comments.

**Technical aspects of the algorithm. (R1, R2)** We chose the parameters $\tau$ (Sinkhorn) and $\gamma$ (learning rate) with grid search prior to running experiments, while $S$ (sampling size) was fixed empirically. In all experiments, we set $\tau = 5, \gamma = 0.2$, and $S = 30$. We set the maximal number of Sinkhorn iterations to 10, and we run stochastic gradient descent for 3000 iterations (even though the algorithm converges long before, after around 1000 iterations, typically). Furthermore, gradient descent was performed in PyTorch with AMSGrad (Reddi *et al.*, ICLR 2018), using automatic differentiation. We did not run the algorithm multiple times to avoid local minima, as the stochastic exploration empirically proved to be very successful in the task. Regarding the sensitivity to different initialisation, the algorithm seems robust in our experiments, and we used random initialisation in all our experiments. Finally, we bypass the computation of the pseudo-inverses by adding a small diagonal shift to the Laplacian matrices and directly computing the inverse matrices, which is orders of magnitude faster. We will add this information to the final version of the paper.

**Convergence of the algorithm and computational complexity. (R1, R2, R3)** Under mild assumptions, GOT algorithm converges almost surely to a local minimum. We do not have any guarantees regarding the global minimum, as the problem is nonconvex. In practice, we observed that the algorithm fails to converge to a good solution only when the compared graphs are very different (we expect to solve this issue with many-to-many correspondences, see next answer). As for the computational complexity, our unoptimized implementation is $O(N^3)$ per iteration. We are currently looking into ways to improve the complexity, by exploiting the sparsity of matrices, and by using fast algorithms to compute the matrix square-root gradient (Lin *et al.*, BMVC 2017).

**Graphs of different sizes (R1, R2).** We will make it clearer in text that our algorithm is aimed at graphs of the same size. In this regard, please note that our method can be extended to graphs of different sizes. We are currently working on the idea of allowing many-to-many correspondences between vertices, which generalises the one-to-one matching considered in our current approach. This will also make it more robust for comparing graphs that cannot be matched perfectly. Given the significant body of additional material, we feel that this topic is best left to a future publication.

**The distance captures the "global structure of graphs" (R1)** Although we do not have a formal proof due to the lack of standard definition for the global structure, our main idea is that graphs can be seen through the distribution of smooth signals. Therefore, the discrepancy in lower graph frequencies, known to capture the global graph structure, will be more important in defining the distance. We supported this claim with the example in Figure 1, where two perturbed copies are compared to the original graph, both with the same amount of removed edges, but a different degree of structural change. The example shows that the proposed distance manages to capture the structural change efficiently.

**Missing references (R2)** We thank the reviewer for suggesting several important works to complement the graph alignment overview. We will gladly include them in the final version. It will be interesting to see whether any of the proposed methods can improve the results of our algorithm, and we will make sure to investigate this in the future. **For directed graphs (R2)** The problem in extending this work to directed graphs indeed lies in the definition of the Laplacian (which is not unique, and it is not positive semi-definite, with possible complex eigenvalues). We do, however, agree it would be very interesting to explore possible extensions of our work to directed graphs.

**Behaviour of the optimal transportation plan (R3)** While it is difficult to derive any guarantees on the behaviour of the transport plan even if the observed signals are smooth, we note that the transport plan is a continuous Lipschitz bounded operator, suggesting some regularity in its behaviour for all signals. That being said, we do agree that signals which are more likely in the observed distribution will have a more robust transportation. An interesting direction would be to derive a transportation plan through a graph filter, a more appropriate Gaussian model for non-smooth signals (Segarra *et al.*, TSIPN 2016). **Use of KL divergence (R3)** In our experiments with KL divergence, numerical issues prevented the algorithm to run until convergence. However, we did not investigate the question thoroughly, and we will rephrase the sentence on line 136-137 to avoid any confusion. Nonetheless, it would be interesting to compare this approach to our method. **Example of Graph Signal Transportation (R3)** The nearest neighbours are computed with the Euclidean distance. The edge weights in the constructed k-NN graph are all equal to 1. The Laplacian matrix is then squared (multiplied by itself) to capture 2-hop distances and create more meaningful weights. The number of neighbours was chosen arbitrarily. We note here that the construction of the graphs is important and a more meaningful graph (obtained through graph inference methods for example) could potentially yield better results. **Comparisons with NetLSD (R3).** We agree that NetLSD is a very interesting method with a different approach that could be used for graph classification. Our preliminary results on the experiment in Figure 5 indicates that it yields 74/100 accuracy. While not as good as some other investigated methods, we note that this is a very good score for a method not taking into account any sort of alignment of nodes. We will add the full comparison to NetLSD in the final version of the paper, including the confusion matrix.

[Meta-Review · NeurIPS 2019]

Many thanks to the authors for their submission. All reviewers have enjoyed this paper and support acceptance. I agree with reviewers that the fact that this only works for now on graphs of the same size is a problem that needs to be fixed. l.124: this is a transport map, not a plan. p.2 would benefit from some rewriting and structuring, maybe using paragraph headers. l.108: references to [26,27] are kept to the minimum. It might be worth spending a little more time on this approach to represent and play with graphs. l.116: it is important to explain that this is a distance between aligned graphs, i.e. distances between sets of edges that share the same vertices and not graphs in full generality. fig.2: right is a bit misleading, since S_\tau(\tilde{P}) seems very close to an assignment and not a bistochastic matrix. l.168: this is not exactly an abuse of notation, but rather a different way to define a covariance matrix. Altough P'LP was a simple operation (reordering of columns), here was is achieved is a mixture of the laplacian matrices lines/columns. Any considerations that is more precise could be useful.